# A Strategy for Selecting “Q-Markers” of Chinese Medical Preparation via Components Transfer Process Analysis with Application to the Quality Control of Shengmai Injection

**DOI:** 10.3390/molecules24091811

**Published:** 2019-05-10

**Authors:** Chunxia Zhao, Huan Liu, Peiqi Miao, Houen Wang, Heshui Yu, Chunhua Wang, Zheng Li

**Affiliations:** 1College of Pharmaceutical Engineering of Traditional Chinese Medicine, Tianjin University of Traditional Chinese Medicine, Tianjin 301617, China; zhaochunxia199411@163.com (C.Z.); 13512039643@163.com (H.L.); miaopeiqi@163.com (P.M.); 13072200650@126.com (H.W.); hs_yu08@163.com (H.Y.); 2Tianjin Key Laboratory of Modern Chinese Medicine, Tianjin University of Traditional Chinese Medicine, Tianjin 301617, China

**Keywords:** components transfer process analysis, quality-marker, Shengmai injection, UPLC-QTOF-MS^E^, fructose

## Abstract

Chinese medical preparation has complicated chemical constituents. Consequently, the proper quality control methods for these Chinese medical preparations have been great challenges to the traditional Chinese medicine modernization and internationalization. What components should be chosen for quality control is a big challenge in the development of traditional Chinese medicine. A new concept of “Quality Marker” was proposed by Liu et al. to solve this problem and established a new research paradigm for traditional Chinese medicine quality study. Several strategies were proposed by the researchers in traditional Chinese medicine, here, we used Shengmai injection as an example to discuss a strategy for selecting “Quality Markers” of Chinese medical preparation by the components transfer process analysis in the Shengmai injection manufacturing process. Firstly, a total of 87 compounds were identified or partially characterized in shengmai injection. Secondly, referenced to the quality control method in China pharmacopeia and considered the biomarkers in the original medicines and representative components in the manufacturing process, four ginsenosides in *Panax ginseng* (Hongshen), two compounds in *Schisandra chinensis* (Wuweizi), and a sugar from *Ophiopogon japonicas* (Maidong) were quantified. As a result, these seven representative compounds exhibited an acceptable transitivity throughout the Shengmai injection manufacturing process. Finally, combined with the active ingredients, components transfer process analysis, and comprehensive evaluation by “Spider-web” analysis, six compounds were selected as the quality markers for the quality control of Shengmai injection. Through this strategy of optimization for quality markers of Shengmai injection, we found that these six compounds could represent the main bioactive substances and be easily detected in the whole process of production. Furthermore, the quality control method was developed for quality assessment and control of these six quality markers in the Shengmai injection. The total content range of the selected quality markers in the 10 batches of the Shengmai injection is 13.844-22.557 mg/mL.

## 1. Introduction

Due to the diversity of chemical components in traditional Chinese medicine (TCM), the complexity of its mechanism of action, and the variability of the preparation process, the quality assurance is more difficult than chemical and biological drugs and necessitated the establishment of a unique quality and bioactivity evaluation system to ensure the safety and effectiveness of TCMs [1,2,3]. 

Professor Liu Changxiao introduced a new concept on the quality marker (Q-marker) for quality assessment and process control of TCMs [4]. The Q-marker is defined as the hundreds of inherent compounds from the herb medicine or other compounds generated by the preparation process, whose biological activity is closely related to their safety and therapeutic effects. It is the chemical substance that is transferable and traceable in the process of production and preparation. The Q-marker is designed for quality assessment and production process control of TCM products with transitivity and traceability [4,5]. Therefore, it is an important job and a great challenge to select appropriate Q-markers for quality assessment and process control of Chinese medical preparation (CMP). Several strategies of choosing Q-markers have been proposed in CMPs. For instance, Li et al. showed that nuciferine and paeoniflorin were identified as promising Q-markers of the Tangzhiqing tablet (TZQ) based on fingerprint analysis, the multi-component quantitative analysis and the dose-exposure-response relationship of TZQ [6]. Jiang et al. constructed a novel strategy of “Spider-web” mode to identify Q-markers by comprehensively integrating “content-stability-pharmacokinetics-pharmacology” of the candidate compounds [7]. 

Based on the core concept of Q-markers, a strategy for quality evaluation based on components transfer process analysis is discussed in this paper. The Shengmai injection (SMI), a TCM formula used for the treatment of cardiovascular and cerebrovascular diseases [8,9,10], consists of three TCMs, i.e., *Panax ginseng* (*P. ginseng*), *Schisandra chinensis* (*S. chinensis*)*,* and *Ophiopogon japonicus* (*O. japonicus*). This study used SMI as an example to discuss a strategy for selecting Q-markers via components transfer process analysis. In this work, ultra-performance liquid chromatography tandem quadrupole time-of-flight mass spectrometry (UPLC-QTOF-MS^E^) was used for the qualitative characterization of the major chemical constituents in SMI. As a result, a total of 87 chemical constituents were identified from SMI on the basis of retention times, accurate masses, mass spectrometric fragmentation characteristic ions. An extract intermediate is an extracted and purified intermediary of the production process, and its quality is crucial to the safety and efficacy of the final product [11]. Through tracking the contents of main chemical constituents (ginsenoside Rg_1_ (Rg_1_), ginsenoside Re (Re), ginsenoside Rb_1_ (Rb_1_), ginsenoside Rd (Rd), schisandrol A (SolA), schisandrol B (SolB), d-Fructose (Fru)) in the SMI manufacturing process by the high-performance liquid chromatography (HPLC) method, The quantitative analysis of these seven compounds were used to confirm that the seven ingredients were transferable.

Based on the results of qualitative analysis and the quantitative determination of compounds, combined with the active ingredients, components transfer process in the SMI manufacturing process and comprehensive evaluation by the “Spider-web” mode, six chemicals were selected as the Q-markers of SMI and UPLC coupled with diode array detection (UPLC-DAD) and HPLC coupled with evaporative light scattering detector (HPLC-ELSD) methods were developed to analyze the contents of Q-markers for quality assessment and control of SMI. The total content range of the selected Q-markers in the 10 batches of SMI is 13.844–22.557 mg/mL. This is the first time that both sugar and fat-soluble components were used as integrative Q-markers in the quality control method of SMI. 

## 2. Results and Discussion

### 2.1. Chemical Characterization of SMI by HPLC–QTOF-MS^E^

Positive and negative ion modes were used to obtain comprehensive data for structural characterization. The representative based peak intensity (BPI) chromatograms of SMI illustrated in Figure 1 and the negative ion BPI chromatogram of reference standards of ginsenosides illustrated in Figure 2. As shown in Appendix A, a total of 87 chemical constituents were identified from the SMI based on fragmentation information, retention times, accurate relative molecular masses, and compounds reported in the related literature and the established chemical composition database of SMI. These compounds were mainly derived from *P. ginseng* and *S. chinensis*, including ginsenosides, lignans, and organic acids, and only one component was identified in *O. japonicus*.

### 2.2. The Study of the Components Transfer Process

*P. ginseng* extract intermediates included the processes of alcohol extraction (four times), first single-effect concentration, static filtration, second single-effect concentration, water precipitation, before sterilization, and after sterilization. *S. chinensis* and *O. japonicus* extract intermediates contained the processes of water extraction (three times), first alcohol precipitation, first single-effect concentration, second alcohol precipitation, second single-effect concentration, water precipitation, before sterilization, and after sterilization (Figure 3). The contents of seven representative compounds in each procedure were quantified by high-performance liquid chromatography photometric diode array (HPLC-PDA) and HPLC-ELSD methods. According to the accumulated data in the process of production, their contents in the raw materials were reverted to the finished product. As shown in Figure 4, Figure 5 and Figure 6, the content of Rd changed slightly, but the content of Rg_1_, Re, and Rb_1_ were reduced in the transfer process and the percentage of variation was nearly 50%. However, SolA and SolB exhibited a downward trend in contents, the process of water precipitation had the greatest influence on SolA and SolB. After the process of water precipitation, the contents of SolA and SolB were also steady and close to zero. The content of Fru also varied slightly after the process of water precipitation. 

### 2.3. Quantitative Analysis of the Major Constituents in SMI by UPLC-DAD and HPLC-ELSD

As shown in Table 1, the linearity results for seven compounds showed good linear correlation (R^2^ > 0.9992) in the range of the test. In the precision test and the stability test, the RSD values of the peak areas of the seven components were less than 0.57% and 1.63%. The results of average recovery test ranged from 96.50% to 101.42% with an RSD less than 2.089%. Experimental results showed that the established UPLC-DAD and HPLC-ELSD methods had a high sensitivity, good precision, satisfactory reproducibility which would be adopted for the quality assessment and control of SMI.

The contents of seven compounds in ten batches of SMI were listed in Table 2 and shown in Figure 7 and Figure 8. The results indicated that larger differences were found between the content of fat-soluble components and the content of water-soluble ingredients. The average content of the components varied slightly in ten batches of SMI samples, with RSD of 4.60–15.06%. Among the seven constituents, the highest amounts (13.30–22.02 mg/mL) of Fru were found in SMI with the lowest amount (0.002–0.003 mg/mL) in SolB. 

### 2.4. Selection of Q-Markers

SMI consisted of three TCMs, including *P. ginseng*, *S. chinensis,* and *O. japonicus*. In the Chinese Pharmacopoeia (2015 edition), the contents of Rg_1_, Re, and Rb_1_ were used for quality control of *P. ginseng*, SolA was used for *S. chinensis*, ruscogenin was used for *O. japonicus*. According to the Chinese Pharmacopoeia report (2015 edition), no representative compounds in *O. japonicus* were included in the content standards of SMI. We studied the components transfer process of methyl radix ophiopogonis flavanone A, methyl radix ophiopogonis flavanone B, and radix ophiopogonis saponins D in *O. japonicus*, but in the processes of water extraction, water precipitation, before sterilization, and after sterilization could not detect these three compounds (Figure 9). Therefore, they were disqualified to be Q-markers. Many studies have shown that carbohydrates had diverse physiological activities [12,13]. Therefore, we considered carbohydrates as representative components of the quality control of *O. japonicus* and its preparations. The content of fructose in *O. japonicus* was much higher than other oligosaccharides. Thus we selected fructose as a representative compound in *O. japonicus* and SMI. As a result, the total content range of the selected Q-markers in the 10 batches of SMI is 13.844–22.557 mg/mL. 

The procedure of water precipitation had the most significant impact on the seven representative components transfer process, which was crucial to the production process control of SMI. Then, in reference to the “Spider-web” mode which was proposed by Jiang et al., we comprehensively evaluated seven representative components to select Q-markers by the dimensions of content level, content consistency in the extract intermediates of raw materials, and the finished product SMI (Figure 10) [7]. We compared the contents and RSD% of seven representative compounds in 10 batches of SMI to obtain the rank value, and calculated the RSD% of seven representative compounds in different processes of four batches of *P. ginseng*, 10 batches of *S. chinensis*, and *O. japonicus*, the average value of RSD% was taken to assess compounds. Regression area (A) of seven representative components in the “Spider-web” mode could objectively reflect the contribution of components in SMI. The bigger regression area suggested the most important contribution. The variables’ coefficient variation was a parameter to reflect the dispersion degree of different variables of the tested compound. The smaller coefficient variation (CV) implied that the compound was qualified to be Q-marker. The importance index (IMI) was employed to discriminate Q-markers, ImI = A × 1/CV, We investigated the seven compounds from three dimensions to obtain IMI. Based on the IMI, the sequence was Rg_1_ > Rb_1_ > Re > Rd > SolA > Fru > SolB (Table 3). SolB had the lowest score in all three dimensions, and it was not recommended as a Q-marker. It had been shown that Rg_1_ could regulate mitochondrial dynamics imbalance via modulation of glutamate dehydrogenase and mitofusin 2 to prevent myocardial hypoxia/reoxygenation injury [14]. Re had many beneficial pharmacological effects on antiarrhythmia and cardiac electrophysiological function [15]. Rd and Rb_1_ contributed to the attenuation of cardiac hypertrophy [16,17]. SolA was able to inhibit cytochrome P450-3A4 activity without altering cellular glutathione level [18]. Low-fructose diet in subjects with chronic kidney disease could reduce inflammation with some potential benefits on blood pressure [19]. According to the pharmacopoeia and related literature along with the active ingredients, components transfer process in the SMI manufacturing process and comprehensive evaluation by “Spider-web” mode, six representative compounds were chosen as the Q-markers, and the UPLC-DAD and HPLC-ELSD methods were developed for the quality assessment and control of SMI. However, these methods also had limitations. Firstly, UPLC-DAD at 203 nm was not the maximum absorption wavelength of SolA and SolB. Secondly, fructose not only existed in *O. japonicus*, but also *P. ginseng* and *S. chinensis*. Although fructose was not a specific component of *O. japonicus*, the content in *O. japonicus* was much higher than *P. ginseng* and *S. chinensis* (Figure 11).

## 3. Methods

### 3.1. Materials and Reagents 

Authentic standards of ginsenoside Rg_1_, ginsenoside Re, ginsenoside Rf, ginsenoside Rb_1_, ginsenoside Rc, ginsenoside Rg_2_, ginsenoside Rh_1_, ginsenoside Ro, ginsenoside Rb_2_ and ginsenoside Rd, schisandrol A, schisandrol B were all purchased from Shanghai yuanye Bio-Technology Co., Ltd. (Shanghai, China). d-Fructose was purchased from National Institutes for Food and Drug Control (Beijing, China). The purity of each chemical was equal to or greater than 97%. They were stored at 4 °C before use. *P. ginseng*, *S. chinensis,* and *O. japonicus* extract intermediates and SMI samples were from SZYY Group Pharmaceutical Limited (Nanjing, China).

All solvents, including methanol and acetonitrile with a purity of 98% were of chromatographic grade purchased from Fisher (Hampton, NH, USA), all other reagents and chemicals were of analytical grade.

### 3.2. Sample Preparation

#### 3.2.1. Preparation of Sample Solutions for UPLC-QTOF-MS^E^ Analysis

A certain amount of reference standards of ginsenoside Rg_1_, ginsenoside Re, ginsenoside Rf, ginsenoside Rb_1_, ginsenoside Rc, ginsenoside Rg_2_, ginsenoside Rh_1_, ginsenoside Ro, ginsenoside Rb_2_ and ginsenoside Rd were mixed to get reference solution. The solution was filtered through a 0.22 μm membrane before UPLC-QTOF-MS^E^ (Waters, Milford, MA, USA) analysis.

The SMI was filtered through a 0.22 µm microporous membrane before UPLC-QTOF-MS^E^ analysis.

#### 3.2.2. Preparation of Sample Solutions for the Study of Components Transfer Process

Reference standards of ginsenoside Rg_1_, ginsenoside Rb_1_, ginsenoside Rd, ginsenoside Re were accurately weighted and prepared by dissolving compounds in 21% acetonitrile-water. The concentrations of ginsenoside Rg_1,_ ginsenoside Rb_1,_ ginsenoside Rd, ginsenoside Re were 499 µg/mL, 503 µg/mL, 303 µg/mL, and 500 µg/mL, respectively. A mixed solution containing all of the four reference compounds were further diluted with 21% acetonitrile-water to obtain four reference solutions with different concentrations. The stock solution was stored at −80 °C. Diluted 2, 4, 8, 16, and 32 times reference solutions were injected at 50 µL; diluted 32 times reference solution was successively injected at 25 µL and 12.5 µL; diluted 64- and 128-times reference solutions were injected at 10 µL to obtain the calibration curves.

In addition to the second single-effect concentrated solution, *P. ginseng* extract intermediates were filtered through a 0.22 µm microporous membrane. 1 mL of second single-effect concentrated solution were transferred to a 5 mL volumetric flask, mixed with a suitable volume of water, and filtered through a 0.22 µm microporous membrane before analysis.

Reference standards of schisandrol A and schisandrol B were accurately weighed and prepared by dissolving compounds in 53% acetonitrile-water. The concentration of schisandrol A was 503 µg/mL and schisandrol B was 299 µg/mL. The solution was further diluted with 53% acetonitrile-water to obtain reference solutions with different concentrations. The stock solution was stored at −80 °C.

The process for preparing *S. chinensis* extract intermediates were similar to that of *P. ginseng* extract intermediates, except that the first single-effect concentrated solution of *S. chinensis* needed to be diluted 5 times. 

The reference standard of d-Fructose was accurately weighed and prepared by dissolving compounds in water. The concentration of d-Fructose was 6.002 mg/mL and diluted with water to obtain six reference solutions with different concentrations. The stock solution was stored at −80 °C.

The solution of first water extraction, first alcohol precipitation, and second alcohol precipitation of *O. japonicus* were diluted 5 times by adding to water. The solution of second water extraction and third water extraction did not need to be diluted, but other processes of the *O. japonicus* intermediates were diluted 50 times. The solution was filtered through a 0.22 µm membrane before analysis.

#### 3.2.3. Preparation of Sample Solutions for Quantification

The stock solution was prepared by weighing 12.5 mg of ginsenoside Rg_1_, 25.01 mg of ginsenoside Rb_1_, 7.47 mg of ginsenoside Rd, 12.18 mg of ginsenoside Re, 4.7 mg of schisandrol A, 4.87 mg of schisandrol B into a 10 mL flask. The stock solution was diluted 2, 4, 8, 16, 32, 64, and 128 times for the construction of calibration curves. The concentration of d-Fructose was 11.998 mg/mL and diluted with water to obtain six reference solutions with different concentrations.

The SMI was filtered through a 0.22 µm microporous membrane before UPLC-DAD analysis and diluted 2-fold with water before HPLC-ELSD (Waters, Milford, MA, USA) analysis.

### 3.3. UPLC-QTOF-MS^E^ Analysis

#### 3.3.1. UPLC-QTOF-MS^E^ Conditions 

The chromatography was performed with a Waters Acquity UPLC BEH C_18_ column (2.1 mm × 100 mm, 1.7 µm; Waters, Milford, MA, USA) and the column temperature was maintained at 50 °C. 1 μL of the sample was used for separation. The separation was achieved using gradient elution with acetonitrile (A) and 0.1% formic acid in water (B) at a flow rate of 0.3 mL/min: 0–12 min, 5–30% (A); 12–20 min, 30–30% (A); 20–26 min, 30–38% (A); 26–29 min, 38–38% (A); 29–31 min, 38–43% (A); 31–34 min, 43–51% (A); 34–40 min, 51–51% (A); 40–44 min, 51–81% (A); 44–46 min, 81–100% (A); 46–48 min, 100–100% (A).

The parameters of the mass spectrometer were set as follows: capillary voltage, 3 kV in negative ion mode and positive ion mode; cone voltage, 40 V; ion source temperature, 120 °C; desolvation temperature, 450 °C; desolvation gas (N_2_) flow rate, 750 L/h; the first range scan, *m*/*z* 100–1600 Da; collision gas, Argon. During low energy scanning, trap collision energy was 4 eV, transfer collision energy was 6 eV, during high energy scanning, trap collision energy was 15 eV, transfer collision energy was 30–50 eV. The mass range was from *m*/*z* 50 to 1500. Leucine-enkephalin (*m*/*z* 556.2771(+)/554.2615(−)) was selected as the lock mass at a concentration of 400 µg/L and flow rate of 5 µL/min 

#### 3.3.2. Establishment of Chemical Composition Database of SMI

According to the domestic and foreign databases CNKI, Pubmed, Science Direct and related literature, the chemical names, molecular formulas, molecular weight and fragment information of SMI and its three herbal medicines were collected. At the same time, the “Waters MassLynx V4.1 SCN901” software (including the exact mass of each element, Waters Corporation, Milford, MA, USA) was used to calculate the possible molecular formula based on the precise relative molecular mass, and the error was less than or equal to 5 ppm.

### 3.4. HPLC-PDA and HPLC-ELSD Analysis 

The HPLC system used for the analysis of extract intermediates, Kromasil 100-5-C_18_ (4.6 mm × 250 mm, 5μm) column was used for the separation of compounds in *P. ginseng* extract intermediates; the temperature of column was set at 30 °C and 10 μL of the sample was loaded onto the column. The mobile phase was composed of water (A ) and acetonitrile (B ) with a gradient elution at the flow rate of 1 mL/min: 0–23 min, 21–21% (B); 23–24 min, 21–32% (B); 24–44 min, 32–33.5% (B); 44–46, 33.5–40% (B); 46–48 min, 40–95% (B); 48–66 min, 95–95% (B). PDA detection wavelengths were set at 203 nm.

The chromatographic conditions for the analysis of *S. chinensis* extract intermediates were similar to those of *P. ginseng*, except that PDA detection wavelengths were set at 254 nm. Mobile phase gradient: 0–12 min, 53–53% (B); 12–13 min, 53–90% (B); 13–20 min, 90–90% (B).

Fructose was quantified after chromatographic separation on a YMC-Pack NH_2_/S-5 μm/12 nm (250 mm × 4.6 mml. D., 5μm, 12 nm) and kept at 30 °C. Isocratic mode was applied using a 79% acetonitrile-water eluent at 1.00 mL/min. Injection volume was 10 μL. The ELSD drift tube was 60 °C, neb heater was 60%, the carrier gas pressure was 35 psi, and the gain was set at 10.

### 3.5. Quantitative Analysis of Representative Compounds in SMI by UPLC-DAD and HPLC-ELSD 

#### 3.5.1. UPLC-DAD and HPLC-ELSD Conditions

UPLC instrument equipped with the Acqutity UPLC^®^BEHC_18_ (2.1 mm × 100 mm, 1.7 μm) column. 3 μL of the sample was loaded onto the column and the temperature of the column was set at 40 °C. The mobile phase was composed of water (A) and acetonitrile (B) with a gradient elution at the flow rate of 0.3 mL/min: 0–3min, 17–17% (B); 3–13min, 17–21% (B); 13–16min, 21–24% (B); 16–16.5min, 24–28% (B); 16.5–35 min, 28–35% (B). DAD detection wavelengths were set at 203 nm.

The difference in chromatographic conditions between the Fru analysis of SMI and *O. japonicus* extract intermediates were the gain of ELSD, the gain was set at 5.

#### 3.5.2. Method Validation

##### Calibration Curves

Linearity was calculated from the relationship between the average peak area and the different concentration of standard solutions. Afterward, based on the calibration curve, the correlation coefficient was calculated.

##### Precision, Stability, Repeatability, and Recovery

To assess the precision of the method, the six replicates of the standard solution were continuously detected and the RSD% calculated.

The repeatability of the method was determined by analyzing the six times of the sample solutions in parallel.

The samples were injected at 0, 2, 4, 8, 10, 12, and 24 h to evaluate the stability of the method and RSD%. The RSD% was calculated based on the measured peak area of the compound.

Six test solutions were prepared by spiking appropriate amounts of standards into the sample solution. The recovery was carried out by analyzing the prepared solutions and calculated by the equations: recovery (%) = (amount found − original amount)/amount spiked × 100%.

## 4. Conclusions

The quality of CMPs is closely related to the manufacturing process of TCM. So, the Q-Markers of CMPs must be screened from the manufacturing process of TCM. In this work, we tracked the changes of the main chemical components in each process of manufacturing, and the chemical composition with key indexes in the production process was found, which included fat-soluble components and sugars. Moreover, these selected compounds had good bioactivities related to effects of SMI. Combined with the “Spider-web” analysis, six high-content components were chosen as the Q-markers of SMI. Based on these six compounds, UPLC-DAD and HPLC-ELSD methods were established for the quality control of SMI. As a result, the total content range of the selected Q-markers in the 10 batches of SMI is 13.844-22.557 mg/mL. In our opinion, compared with the method previously reported by the literature [6,7], this strategy has better practicability and could be utilized for selecting Q-markers of CMPs, and it might also be deemed as a plausible method for the quantitative analysis of Q-markers in other CMPs.

## Figures and Tables

**Figure 1 molecules-24-01811-f001:**
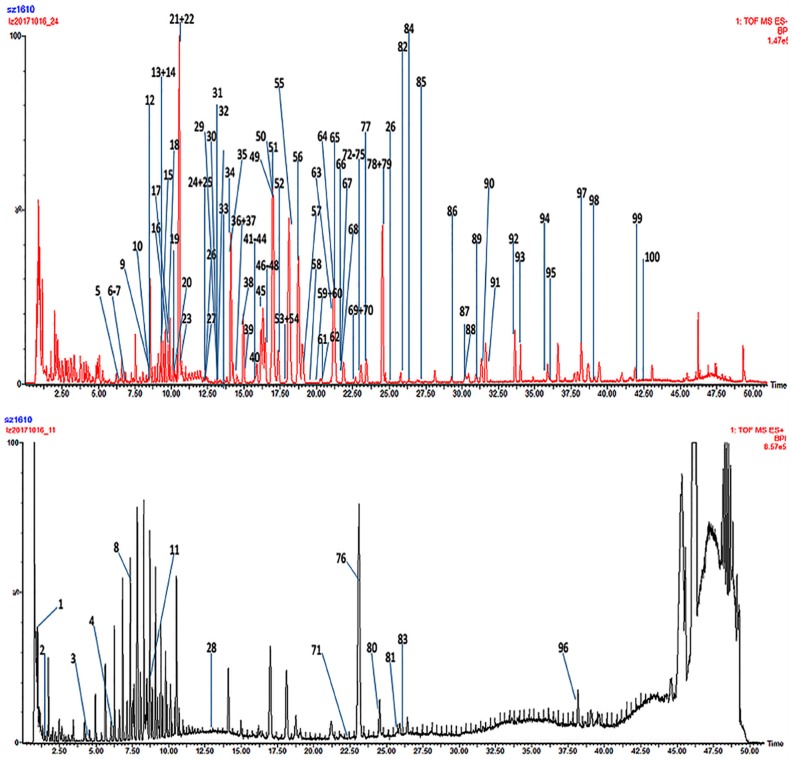
The representative based peak intensity (BPI) total ion chromatogram from a sample of Shengmai injection (SMI) based on ultra-performance liquid chromatography tandem quadrupole time-of-flight mass spectrometry (UPLC-QTOF-MS^E^).

**Figure 2 molecules-24-01811-f002:**
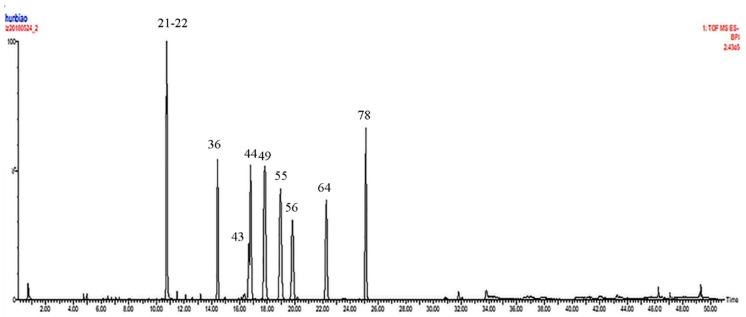
The negative ion BPI chromatogram from a mixed solution of reference standards of ginsenosides based on UPLC-QTOF-MS^E^.

**Figure 3 molecules-24-01811-f003:**
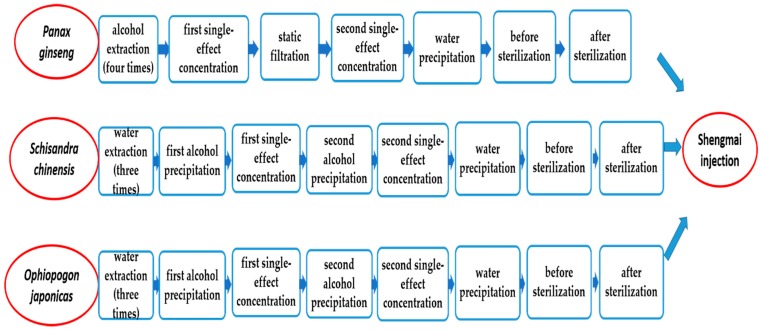
The process flowchart of SMI manufacture.

**Figure 4 molecules-24-01811-f004:**
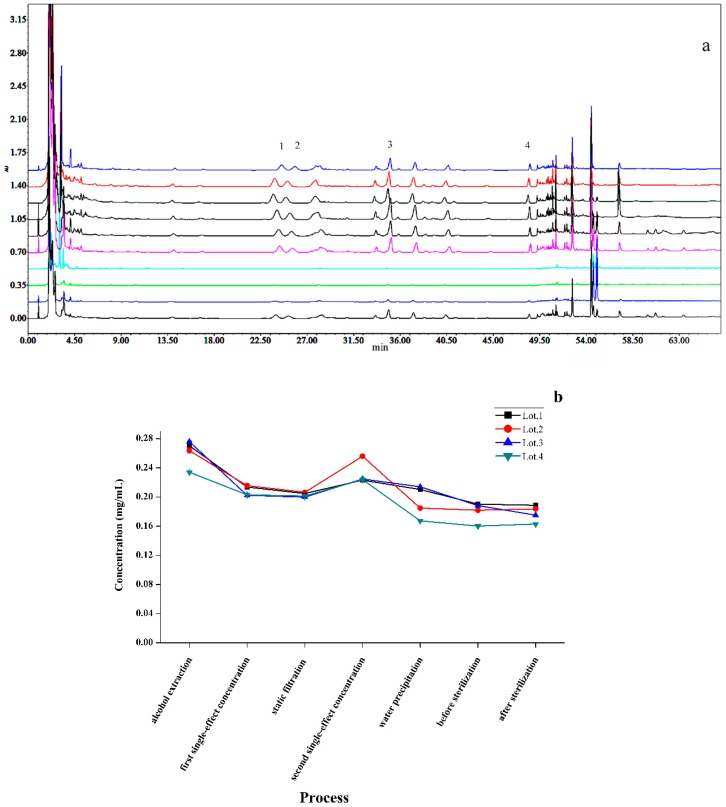
High-performance liquid chromatography photometric diode array (HPLC-PDA) chromatograms of (**a**) the sample solution of *Panax ginseng* extract intermediates including the processes of alcohol extraction (four times), first single-effect concentration, static filtration, second single-effect concentration, water precipitation, before sterilization, and after sterilization (1: Rg_1_, 2: Re, 3: Rb_1_, 4: Rd); the transfer process of (**b**) Rg_1_, (**c**) Re (**d**) Rb_1_, (**e**) Rd.

**Figure 5 molecules-24-01811-f005:**
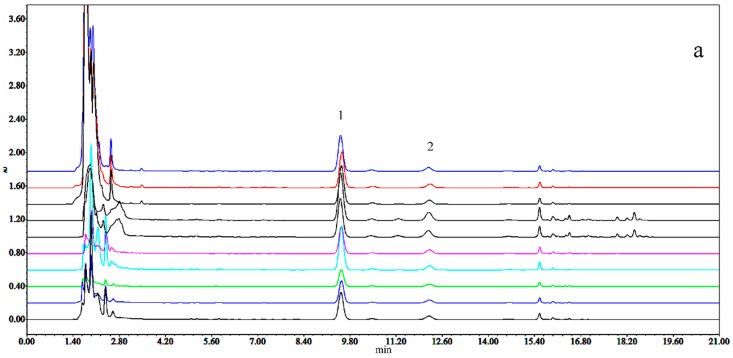
HPLC-PDA chromatograms of (**a**) the sample solution of *Schisandra chinensis* extract intermediates including the processes of water extraction (three times), first alcohol precipitation, first single-effect concentration, second alcohol precipitation, second single-effect concentration, water precipitation, before sterilization, and after sterilization (1: SolA, 2: SolB). The transfer process of (**b**) SolA, (**c**) SolB.

**Figure 6 molecules-24-01811-f006:**
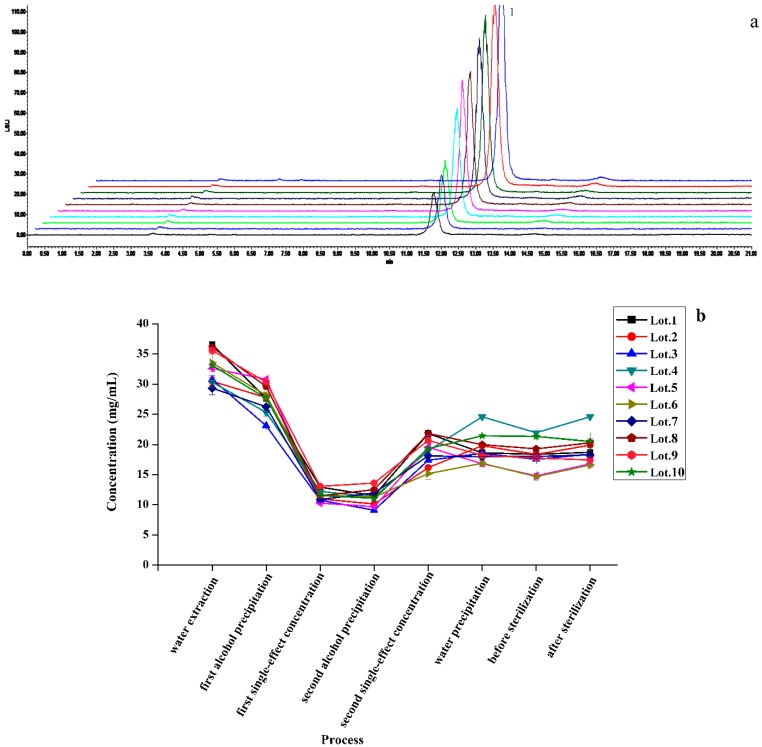
HPLC coupled with evaporative light scattering detector (HPLC-ELSD) chromatograms of (**a**) the sample solution of *Ophiopogon japonicus* extract intermediates including the processes of water extraction (three times), first alcohol precipitation, first single-effect concentration, second alcohol precipitation, second single-effect concentration, water precipitation, before sterilization, and after sterilization (1: Fru). The transfer process of (**b**) Fru.

**Figure 7 molecules-24-01811-f007:**
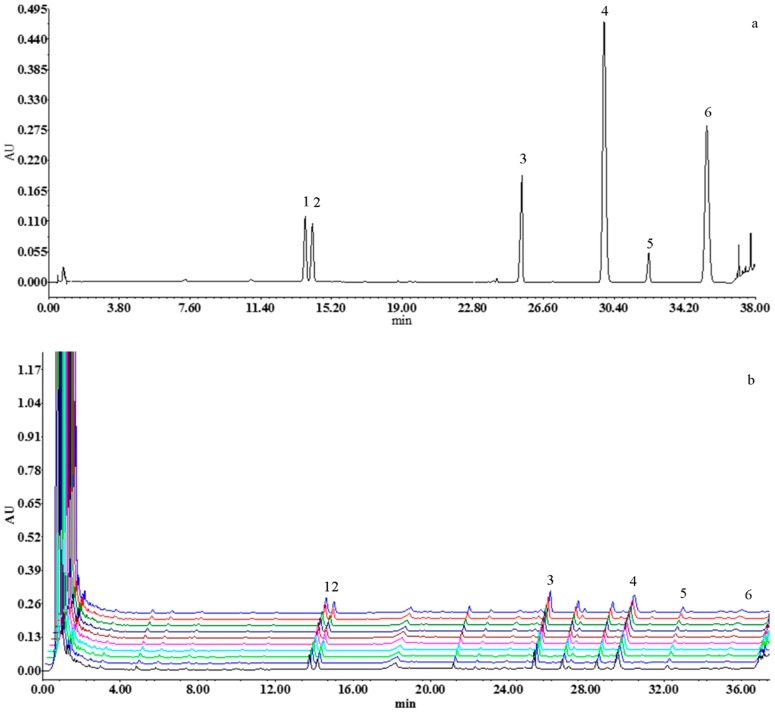
Ultra-performance liquid chromatography diode array detection (UPLC-DAD) chromatograms of (**a**) the mixed reference solution, (**b**) the fingerprint of ten batches of SMI (203 nm, 1: Rg_1_, 2: Re, 3: Rb_1_, 4: Rd, 5: SolA, 6: SolB).

**Figure 8 molecules-24-01811-f008:**
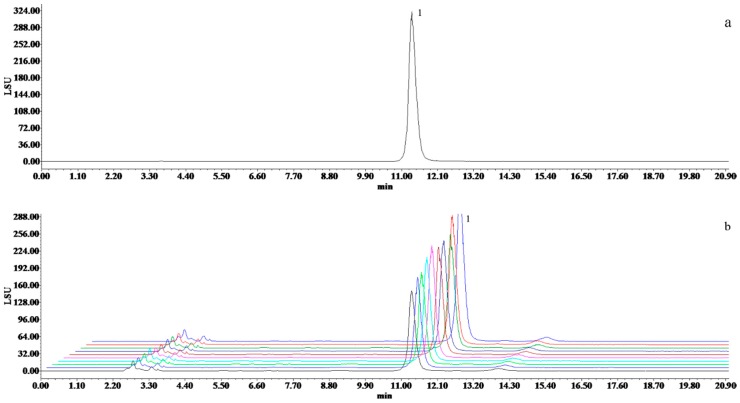
HPLC-ELSD chromatograms of (**a**) the reference solution of fructose, (**b**) the fingerprint of 10 batches of SMI (1: Fru).

**Figure 9 molecules-24-01811-f009:**
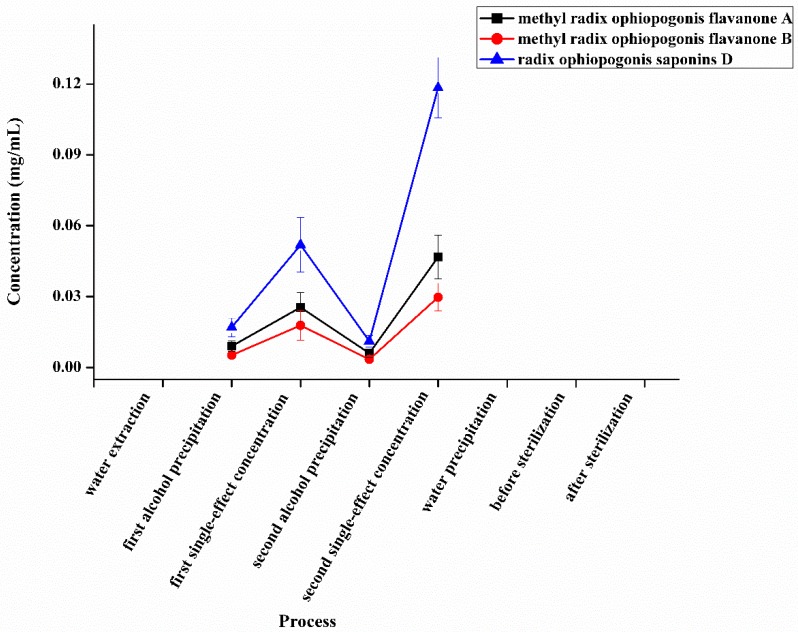
The transfer process of methyl radix ophiopogonis flavanone A, methyl radix ophiopogonis flavanone B, and radix ophiopogonis saponins D in *O. japonicus*.

**Figure 10 molecules-24-01811-f010:**
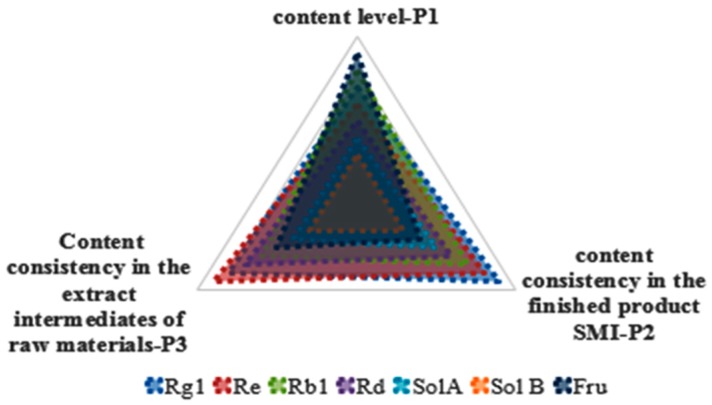
A comprehensive evaluation of the seven representative compounds based on the “Spider-web” mode.

**Figure 11 molecules-24-01811-f011:**
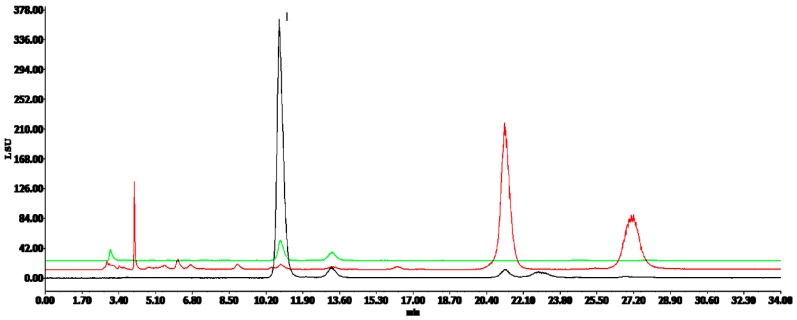
HPLC-ELSD chromatograms of *P. ginseng* primary water extracts, *S. chinensis* primary water extracts, and *O. japonicus* primary water extracts. (1: Fru).

**Table 1 molecules-24-01811-t001:** Regression equation, R^2^, linear range (mg/mL), precision, repeatability, stability, recovery for 7 compounds.

Compounds	Regression Equation	R^2^	Linear Range (mg/mL)	Precision (RSD%)	Repeatability (RSD%)	Stability (RSD%)	Average Recovery (%)	Recovery (RSD%)	LOQs (mg/mL)	LODs (mg/mL)
Rg_1_	Y = 3.306 × 10^6^X − 4240.81	0.9999	0.00990–0.317	0.57	0.83	0.27	101.42	1.738	1.22 × 10^−3^	4.07 × 10^−4^
Re	Y = 2.976 × 10^6^X + 2341.61	0.9997	0.00952–0.305	0.53	1.36	0.21	97.46	2.050	1.16 × 10^−3^	3.87 × 10^−4^
Rb_1_	Y = 2.619 × 10^6^X − 1101.56	0.9998	0.0195–0.625	0.46	0.60	0.16	100.33	1.933	1.35 × 10^−3^	4.48 × 10^−4^
Rd	Y = 2.751 × 10^6^X + 5171.96	0.9992	0.00584–0.187	0.44	1.50	0.26	100.48	2.062	1.30 × 10^−3^	4.33 × 10^−4^
SolA	Y = 5.523 × 10^7^X + 36309.46	0.9998	0.00347–0.111	0.45	0.63	0.43	100.89	2.089	5.52 × 10^−6^	1.84 × 10^−6^
SolB	Y = 5.505 × 10^7^X + 13971.00	0.9995	0.00179–0.0573	0.43	2.65	1.63	96.50	1.041	7.36 × 10^−6^	2.45 × 10^−6^
Fru	lgY = 1.3943lgX + 5.21081	R^2^ = 0.9994	0.74988–11.998	0.25	0.18	0.30	97.18	2.06	0.119	0.0595

**Table 2 molecules-24-01811-t002:** Contents of the seven compounds in the 10 batches.

NO.	Contents (mg/mL)						
	Rg_1_	Re	Rb_1_	Rd	SolA	SolB	Fru
Lot.1	0.135	0.101	0.197	0.052	0.015	0.002	14.36
Lot.2	0.149	0.111	0.212	0.056	0.016	0.002	13.30
Lot.3	0.137	0.102	0.199	0.052	0.015	0.002	17.47
Lot.4	0.137	0.102	0.197	0.052	0.015	0.002	16.83
Lot.5	0.153	0.117	0.232	0.061	0.016	0.002	15.15
Lot.6	0.14	0.103	0.196	0.053	0.02	0.003	16.68
Lot.7	0.143	0.106	0.182	0.046	0.016	0.002	19.47
Lot.8	0.137	0.103	0.195	0.053	0.015	0.002	17.39
Lot.9	0.149	0.111	0.21	0.051	0.016	0.002	22.02
Lot.10	0.136	0.102	0.193	0.048	0.016	0.002	17.61
Average	0.142	0.106	0.201	0.052	0.016	0.002	17.03
RSD%	4.60	5.10	6.80	7.80	9.32	15.06	14.68

**Table 3 molecules-24-01811-t003:** The ranking value and score value of the three dimensions, the IMI, and ranking of the compounds.

Compounds	P1		P2		P3		IMI	Final Rv
	Rv	Sv	Rv	Sv	Rv	Sv		
Rg_1_	3	0.7	1	0.9	2	0.8	19.20	1
Re	4	0.6	2	0.8	1	0.9	11.54	3
Rb_1_	2	0.8	3	0.7	4	0.6	14.70	2
Rd	5	0.5	4	0.6	3	0.7	10.80	4
SolA	6	0.4	5	0.5	6	0.4	9.76	5
SolB	7	0.3	7	0.3	7	0.3	0.12	7
Fru	1	0.9	6	0.4	5	0.5	4.08	6

Note: Rv: rank value, Sv: Score value, IMI: the importance of index, Final Rv: final rank value.

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
