# Peer review of "A Strategy for Selecting “Q-Markers” of Chinese Medical Preparation via Components Transfer Process Analysis with Application to the Quality Control of Shengmai Injection"

_molecules, 2019, doi:10.3390/molecules24091811_

Round 1

Reviewer 1 Report

The paper reports a strategy for selecting quality markers of a Chinese medical preparation, namely, Shengmai injection, by means of the component transfer process analysis in order to apply them to the quality control of this kind of preparations. The authors identify all the possible constituents in Shengmai injection by UPLC-QTOF-MS/MS, use HPLC to quantify the most relevant compounds, coming from three different traditional Chinese medicine, and finally, they apply Spider-web tools to evaluate and select the quality markers in terms of several dimensions, such as content level, content consistency and the final Shengmai injection product. The authors conclude the possible applicability of this methodology to the quantitative analysis of quality markers in other Chinese medical preparations.

The paper seems to be interesting. However, novelty of the manuscript is not clear or not well stated; besides, several details need to be corrected and much improvement needs to be done.

General remarks

English grammar and spelling corrections should be done throughout the whole text. Please, revise carefully the text. Some parts of some sentences are repetitive (i.e., lines 68-69) or there are some words missing, giving no sense to the statement (i.e., lines 79-82). Please, check the text.

Revise acronyms in the text. I suggest defining the acronyms at the beginning of each section, or at least in the Introduction section, not only in the abstract.

Better resolution and bigger images should be provided in general. Moreover, a legend could be added to the chromatograms to explain the different colors. Captions of figures should have more experimental details.

Specific remarks

1) Abstract section: This section needs to be reformulated to explain well the core and novelty of the paper. Many concepts regarding traditional Chinese medicine and so on are supposed to be known previously by readers, and it should not be in this way.

2) Introduction section: in this part, hypothesis, novelty and importance of the work reported should be stressed in an appropriate way. Some pieces of information should be changed to other sections, such as Results and Discussion or Conclusions sections. The last part is very similar to Abstract, and hence, too much repetitive. Thus, much improvement needs to be done in this section.

3) Results and Discussion section: in general, this section shows an important lack of continuity. The different parts should be reorganized to present the results and the discussion much more coherently. Perhaps some sentences introducing and linking each part could solve out this problem, apart from reorganization.

More discussion could be also added to the different Figures and Tables reported. Table S1 is not cited in the text or discussed in a general manner at least. Connection of this Table with the results and discussion should be done. Figure 1 is not cited in the text or discussed either.

Subsection 2.2: It is supposed that at the beginning of this section the process described corresponds to SMI manufacture. Perhaps a better explanation together with a block diagram/scheme could be provided for clarification purposes.

In line 114, the word ‘slightly’ maybe serves for e) plot, but for c), b) and d) plots, percentage of variation are in the order of 50% or more. Please, check.

At the end of sentence in line 117, I would include: ‘… SolA and SolB were also steady and close to zero’.

In Figure 3, 4 and 5, what do the different colors mean, either in the chromatograms or in the other plots? Some legend should be added. In the corresponding captions of these figures, the legend of numbers 1, 2, 3 and 4, corresponding to the different compounds should be put just after the chromatogram. In general in these figures, scales and axis cannot be seen properly.

Why do the authors report the results of HPLC in two tables? Why not in only one table? The same can be said for Tables 3 and 4. It should be more appropriate. Besides, consider significant ciphers in general in all the tables reported.

The limit of detection for SolA and SolB are much lower than for the other compounds and for Fru is too much higher. Do they have an explanation for these facts?

Figure 6 is not cited or explained in the text. In caption of Figure 7, numbers (1 to 6) are not explained.

I think the subsection 2.4 should appear unless appropriate explanation is given for being at this position (this suggestion is related to the reorganization of Results and Discussion section).

Add some legend to Figure 9 for explaining colors.

Paragraph from lines 190 to lines 192 should appear in other part of the Results and Discussion section or even in the Introduction section or after considering these compounds as the most important ones for quality marker assessment.

Interpretation of Figure 10 should be given and Table 5 should have also a legend for explaining the meaning of each variable.

4) Methods section: full names of the most important compounds should be used here and the first time they are named in the text (Introduction and Results and Discussion section, apart from Methods section). Purity of reagents should be also included. The first time an apparatus is mentioned, all the information concerning the instrumentation (brand, country, and so on) should be mentioned (i.e., UPLC is named for the first time in section 3.2.1, but information of this instrumentation is given in section 3.5.1). Please try to reorder the information here as well.

5) Conclusions: Some sentences are repetitive and could be removed. In the last sentence of the conclusions, a ‘method previously reported’ is generally mentioned. Please, specify here. Advantages of the methodology proposed should be better stressed.

Author Response

According to the comments, we have revised and highlighted the relevant part in the revised manuscript.

Response to reviewer1 Comments

The paper reports a strategy for selecting quality markers of a Chinese medical preparation, namely, Shengmai injection, by means of the component transfer process analysis in order to apply them to the quality control of this kind of preparations. The authors identify all the possible constituents in Shengmai injection by UPLC-QTOF-MS/MS, use HPLC to quantify the most relevant compounds, coming from three different traditional Chinese medicine, and finally, they apply Spider-web tools to evaluate and select the quality markers in terms of several dimensions, such as content level, content consistency and the final Shengmai injection product. The authors conclude the possible applicability of this methodology to the quantitative analysis of quality markers in other Chinese medical preparations.

The paper seems to be interesting. However, novelty of the manuscript is not clear or not well stated; besides, several details need to be corrected and much improvement needs to be done.

General remarks

Point 1:English grammar and spelling corrections should be done throughout the whole text. Please, revise carefully the text. Some parts of some sentences are repetitive (i.e., lines 68-69) or there are some words missing, giving no sense to the statement (i.e., lines 79-82). Please, check the text.

Respond 1:

     We check and modify the error of English grammar and spelling in the text. We delete the duplicate sentence and modify the statement in the revised manuscript (lines 75-79)

Point 2:Revise acronyms in the text. I suggest defining the acronyms at the beginning of each section, or at least in the Introduction section, not only in the abstract.

Respond 2:

We give the full name in the abstract and define the acronyms in the Introduction section.

Point 3: Better resolution and bigger images should be provided in general. Moreover, a legend could be added to the chromatograms to explain the different colors. Captions of figures should have more experimental details.

Respond 3:

We provide better resolution and bigger images in the revised manuscript and add the legend to explain the different colors in chromatograms and include more experimental details in captions of figures.

Specific remarks

Point 4:1) Abstract section: This section needs to be reformulated to explain well the core and novelty of the paper. Many concepts regarding traditional Chinese medicine and so on are supposed to be known previously by readers, and it should not be in this way.

Respond 4:

We modify the section of abstract to explain the core and novelty of the paper.

Point 5:2) Introduction section: in this part, hypothesis, novelty and importance of the work reported should be stressed in an appropriate way. Some pieces of information should be changed to other sections, such as Results and Discussion or Conclusions sections. The last part is very similar to Abstract, and hence, too much repetitive. Thus, much improvement needs to be done in this section.

Respond 5:

We modify the introduction section, remove the repetitive part and stress the importance of the work in the revised manuscript.

Point 6:3) Results and Discussion section: in general, this section shows an important lack of continuity. The different parts should be reorganized to present the results and the discussion much more coherently. Perhaps some sentences introducing and linking each part could solve out this problem, apart from reorganization.

Respond 6:

We reorganize the results and discussion section to increase the continuity of the text and to present present the results and the discussion much more coherently.

Point 7:More discussion could be also added to the different Figures and Tables reported. Table S1 is not cited in the text or discussed in a general manner at least. Connection of this Table with the results and discussion should be done. Figure 1 is not cited in the text or discussed either.

Respond 7:

  We add more discussion to explain the different Figures and Tables reported, Table S1 is cited in the revised manuscript (line 91) and Figure 1 is cited and discuss in the revised manuscript (line 89-95)

Point 8:Subsection 2.2: It is supposed that at the beginning of this section the process described corresponds to SMI manufacture. Perhaps a better explanation together with a block diagram/scheme could be provided for clarification purposes.

 Respond 8:

 As shown in figure 3 in the revised manuscript, we add the process flowchart of shengmai injection manufacture at the beginning of section 2.2.

Point 9:In line 114, the word ‘slightly’ maybe serves for e) plot, but for c), b) and d) plots, percentage of variation are in the order of 50% or more. Please, check.

 Respond 9:

 We modify the description of c), b) and d) plots in the revised manuscript (112-113).

Point 10:At the end of sentence in line 117, I would include: ‘… SolA and SolB were also steady and close to zero’.

Respond 10:

 We modify to SolA and SolB were also steady and close to zero (line 116).

Point 11:In Figure 3, 4 and 5, what do the different colors mean, either in the chromatograms or in the other plots? Some legend should be added. In the corresponding captions of these figures, the legend of numbers 1, 2, 3 and 4, corresponding to the different compounds should be put just after the chromatogram. In general in these figures, scales and axis cannot be seen properly.

Respond 11:

 We add some legend to explain the different colors mean in In Figure 3, 4 and 5. The legend of numbers 1, 2, 3 and 4, corresponding to the different compounds put after the chromatogram and modify the scales and axis in figures.

Point 12:Why do the authors report the results of HPLC in two tables? Why not in only one table? The same can be said for Tables 3 and 4. It should be more appropriate. Besides, consider significant ciphers in general in all the tables reported.

Respond 12:

 We modify Table 1, 2, Table 3, 4 and put in one table, respectively.

Point 13:The limit of detection for SolA and SolB are much lower than for the other compounds and for Fru is too much higher. Do they have an explanation for these facts?

Respond 13:

The reference compounds of SolA and SolB are diluted as far as possible and the limit of detection for SolA and SolB are calculated according to the signal-noise ratio equal to 3:1, because the peak shape of SolA and SolB are more acute than others, the limit of detection for SolA and SolB are much lower than for the other compounds. Fru is detected by general detector, evaporative light scattering detector. The baseline noise of evaporative light scattering detector is much larger than the UV detector, so the limit of detection for Fru is too much higher.

Point 14:Figure 6 is not cited or explained in the text. In caption of Figure 7, numbers (1 to 6) are not explained.

Respond 14:

Bacause the fingerprint of ten batches of SMI (Figure 7) include the chromatograms of Figure 6. So, Figure 6 is deleted in the revised manuscript.We add the legend to explain the numbers in Figure 7

Point 15:I think the subsection 2.4 should appear unless appropriate explanation is given for being at this position (this suggestion is related to the reorganization of Results and Discussion section).

Respond 15:

Thank you, We consider to remove the subsection 2.4.

Point 16:Add some legend to Figure 9 for explaining colors.

Respond 16:

 We add some legend to explain different colors in Figure 9.

Point 17:Paragraph from lines 190 to lines 192 should appear in other part of the Results and Discussion section or even in the Introduction section or after considering these compounds as the most important ones for quality marker assessment.

Respond 17:

 After considering these compounds as the most important ones for quality marker assessment, the revised manuscript appear the paragraph from lines 190 to lines 192.

Point 18:Interpretation of Figure 10 should be given and Table 5 should have also a legend for explaining the meaning of each variable.

Respond 18:

 We add the interpretation of Figure 10 in the revised manuscript and add a note to explain the meaning of each variable in Table 5.

Point 19:4) Methods section: full names of the most important compounds should be used here and the first time they are named in the text (Introduction and Results and Discussion section, apart from Methods section). Purity of reagents should be also included. The first time an apparatus is mentioned, all the information concerning the instrumentation (brand, country, and so on) should be mentioned (i.e., UPLC is named for the first time in section 3.2.1, but information of this instrumentation is given in section 3.5.1). Please try to reorder the information here as well.

Respond 19:

 We modify the full names of the most important compounds in methods section and add the purity of reagents. We reorder the information of instrumentation.

Point 20:5) Conclusions: Some sentences are repetitive and could be removed. In the last sentence of the conclusions, a ‘method previously reported’ is generally mentioned. Please, specify here. Advantages of the methodology proposed should be better stressed.

Respond 20:

We remove the repetitive sentences and stress the advantages of the methodology proposed.

Reviewer 2 Report

Figure 3b, 3c, 3d, 3e, 4b, 4c, 5b and 9 should contain the error bars.

Author Response

Response to reviewer2 Comments

Point 1:Figure 3b, 3c, 3d, 3e, 4b, 4c, 5b and 9 should contain the error bars.

Respond 1:

    We add the error bars in Figure 3b, 3c, 3d, 3e, 4b, 4c, 5b and 9, but the standard deviation is small, it cannot be seen clearly in some Figures.

Round 2

Reviewer 1 Report

Most of the suggestions/comments made by the reviewer have been accomplished. However, Figure 2 is still missing in the text (it is not cited or explained) and in Conclusions section, in the last sentence, the sentence ‘compared with the method previously reported’ is not clarified yet. This sentence is still confusing. After completing these two explanations, the paper can be published in the journal Molecules.

Author Response

According to the comments, we have revised and highlighted the relevant part in the revised manuscript.

Response to reviewer1 Comments

Point 1:Most of the suggestions/comments made by the reviewer have been accomplished. However, Figure 2 is still missing in the text (it is not cited or explained)

Responds:

We cite the Figure 2 in the text.

Point2: in Conclusions section, in the last sentence, the sentence ‘compared with the method previously reported’ is not clarified yet. This sentence is still confusing. After completing these two explanations, the paper can be published in the journal Molecules.

Responds:

We modify the sentence in conclusions section.

Reviewer 2 Report

No extra comment.

Author Response

Response to reviewer 2 Comments

Point : No extra comment

Responds:

Thanks for your comments.